# Outcomes of Hip Reconstruction for Metastatic Acetabular Lesions: A Scoping Review of the Literature

Sandeep Krishan Nayar [1,*] , Thomas A. Kostakos [2], Olga Savvidou [3], Konstantinos Vlasis [3] and Panayiotis J. Papagelopoulos [3]

1   Barts Health NHS Trust, London E1 1BB, UK
2   Whittington Health NHS Trust, London N19 5NF, UK; thomas.kostakos@nhs.net
3   Department of Anatomy, National and Kapodistrian University of Athens, 157 72 Athens, Greece; olgasavvidou@gmail.com (O.S.); kostasvlasis@gmail.com (K.V.); pjporthopedic@gmail.com (P.J.P.)
*   Correspondence: sandeep.nayar@doctors.org.uk

**Abstract:** (1) Background: Management of metastatic periacetabular lesions remains a challenging area of orthopaedics. This study aims to evaluate and summarize the currently available reconstructive modalities, including their indications and outcomes. (2) Methods: A scoping review was carried out in accordance with PRISMA guidelines. Medline, EMBASE, and Cochrane were searched for relevant articles. (3) Results: A total of 18 papers met inclusion criteria encompassing 875 patients. The most common primary malignancy was breast ($n = 230$, 26.3%). Reconstruction modalities used were total hip arthroplasty ($n = 432$, 49.1%), the Harrington procedure ($n = 374$, 42.5%), modular hemipelvic endoprotheses ($n = 63$, 7.2%) and a reverse ice-cream cone prosthesis ($n = 11$, 1.25%). (4) Conclusions: Advances in implant design including use of dual mobility or flanged cups, tantalum implants, and modular hemipelvic endoprostheses allow for larger acetabular defects to be addressed with improved patient outcomes. This armamentarium of reconstruction options allows for tailoring of the procedure performed depending on patient factors and extent of periacetabular disease.

**Keywords:** hip reconstruction; metastatic periacetabular cancer; Harrington procedure

## 1. Introduction

The pelvis is the second most common site for bone metastases after the spine, with patients often presenting with severe pain and reduced quality of life [1,2]. Lesions affecting weightbearing areas of the acetabulum, extensive bone loss, pathological fractures, and acetabular protrusion warrant surgical reconstruction [3]. Goals of surgery are to reduce pain and provide a stable, well-fixed construct which allows for early full weight-bearing and long-term survivorship beyond the expected prognosis of the patient [4].

The surgical approach and technique are guided by the extent of the disease with a myriad of difficulties faced when performing reconstructive surgery. These include the need for large-volume resection and the presence of the surrounding anatomy, including neurovascular and visceral structures.

In 1981, Harrington classified periacetabular lesions secondary to metastatic disease into four classes based on the location and extent of periacetabular bone loss. Furthermore, he described a technique to address extensive defects extending across the anterior and posterior columns not amenable to primary arthroplasty [5]. The reconstructive technique described used threaded Steinmann pins inserted retrograde through the acetabular roof into the iliac wing followed by cement augmentation and the insertion of an acetabular shell with a polyethylene liner. Since Harrington's original paper, a number of advances including imaging modalities, techniques, and implants have been introduced in order to improve outcomes [3]. Despite this, management of such lesions remains a challenging area of orthopaedics with no established best practice [6].

This scoping review aims to evaluate and summarize the currently available reconstructive modalities for metastatic periacetabular lesions, including their indications and outcomes.

## 2. Materials and Methods

### 2.1. Protocol and Registration

A scoping review of the published literature was carried out in accordance with Preferred Reporting Items for Systematic Reviews and Meta-Analyses—Extension for Scoping Reviews (PRISMA-ScR) guidelines [7].

### 2.2. Eligibility Criteria

Inclusion criteria were adults with periacetabular cancer secondary to metastatic disease where instrumented reconstruction was performed. Exclusion criteria were adults with primary malignancy, allograft reconstruction, paediatric patient cohorts (under 18 years), reviews, conference abstracts, opinion-based reports, studies published prior to 2008, and articles not published in English.

### 2.3. Search Strategy

A comprehensive search of the published literature on MEDLINE, EMBASE, and Cochrane from inception to 8 June 2021 was carried out. The following search terms were used: (acetab* OR periacetab* OR peri-acetab* OR hemipelv*) AND (reconstruc* OR replac*) AND (tum*OR metast* OR resec* OR onc* OR canc*).

### 2.4. Selection Process

Two reviewers (SKN and TAK) independently performed eligibility assessment of the articles. This was initially carried out through screening of the article titles followed by abstracts. The process was completed by full-text evaluation. Disagreements between reviewers were resolved via consensus with the senior author (PJP).

### 2.5. Data Collection Process and Data Items

Each included article was reviewed by SKN and TAK. Data items were collected onto Microsoft Excel. These included: demographic information, diagnosis, use of adjuvant therapy, resection margin, location of pelvic resection, implant details, follow up, implant survivorship defined by revision rate, complications, postoperative function in accordance with the Musculoskeletal Tumor Society (MSTS) score and patient survival.

### 2.6. Synthesis of Results

Mean values and ranges were used to describe continuous variables, and percentages for categorical variables. Narrative synthesis of the results was carried out.

### 2.7. Quality Assessment

Risk of bias was assessed for each study by two independent reviewers (SN and TK) in accordance with the Cochrane Collaboration guidelines by using the Risk Of Bias In Non-randomized Studies of Interventions (ROBINS-I) tool. This scores observational studies across seven domains (confounding, participant selection, classification of interventions, deviations from the intended intervention, missing data, measurement of outcomes, and selection of reported results) giving an overall judgement of "low risk", "moderate risk", "serious risk", or "critical risk" of bias.

## 3. Results

### 3.1. Study Selection

A total of 1786 papers were identified from the initial search strategy from which 1020 were duplicates. A total of 766 papers underwent screening of titles and abstracts, of which 683 did not meet inclusion criteria and were thus excluded. The remaining 83 articles were retrieved for full-text review. From these, 18 papers met the inclusion criteria and were

included in this scoping review. The PRISMA flow diagram of study selection is illustrated in Figure 1.

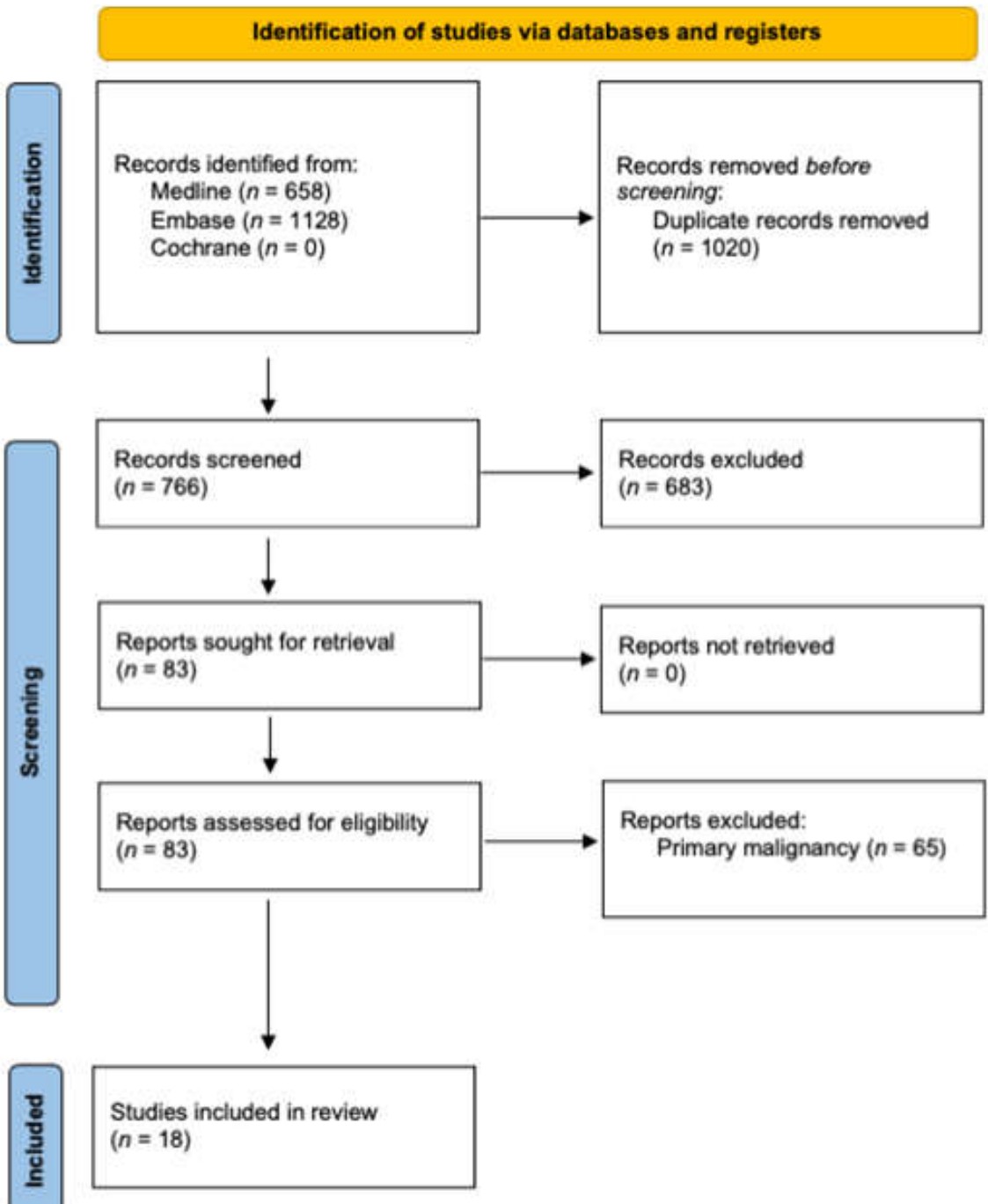

**Figure 1.** PRISMA flow diagram.

### 3.2. Study Characteristics

All the included studies were case series or cohort studies. The number of patients in each study ranged from 9 to 126, with a mean of 49 patients. A total of 875 patients with 883 hip reconstructions were included in this review. The mean age of patients ranged from 51 to 67 years. A total of 476 patients (54%) were female. Follow up ranged from 1 to 205 months, with a mean of 29.6 months.

Study characteristics are summarized in Table 1. Complications according to reconstruction type are summarized in Table 2.

**Table 1.** Summary of studies.

| Study (Country) | Total no. Patients (Hips), *n* | Mean Age, Years (Range) | Harrington Class (*n*) | Reconstruction Technique (*n*) | Adjuvant Therapy (*n*) | Post op MSTS, % (Range) | Implant Survival | Mean Follow up, Months (Range) | Patient Survival Rate |
|---|---|---|---|---|---|---|---|---|---|
| Wei et al., 2021 (China) [8] | 78 (78) | 56 (17–79) | Phase 1: III (*n* = 24) Phase 2: IIIa (*n* = 6) IIIb: (*n* = 48) | Modified Harrington (*n* = 30) Modular hemipelvic endoprosthesis (*n* = 48) | Unknown | Phase 1: 56.5 (20–90) Phase 2: 65.3 (23–97) | Unknown | 18 (12–60) | 50.3% at 2 years |
| Gusho et al., 2020 (USA) [9] | 9 (9) | 60 (40–81) | I (*n* = 1) II (*n* = 2) III (*n* = 6) | Modified Harrington (*n* = 9) | PrR (*n* = 4) PrR + PoR (*n* = 2) PoC (*n* = 1) PrC + PoC (*n* = 5) | 59 (30–83) | 100% at 6 months | 9 (1–14) | 66% at 3 months |
| Houdek et al., 2020 (USA) [10] | 115 (115) | 57 (28–73) | I (*n* = 35) II (*n* = 19) III (*n* = 61) | Modified Harrington (*n* = 78) THA with tantalum acetabular component (*n* = 37) | PrR (*n* = 64) | Unknown | 87% in Harrington & 92% in Tantalum at mean 14 months (2 weeks–7 years) | 48 (12–180) | 34% at 2 years 16% at 5 years 15% at 10 years |
| Kask et al., 2020 (Finland) [11] | 89 (89) | 67 (27–94) | I (*n* = 36) II (*n* = 41) III (*n* = 12) | Modified Harrington (*n* = 89) | PrR (*n* = 31) PoR (*n* = 37) PrR + PoR (*n* = 2) | Unknown | 96% at 1, 2 & 5 years | 18 (12–60) | 46% at 1 year 25% at 2 years 16% at 5 years |
| Houdek et al., 2020 (USA) [12] | 58 (58) | 62 (22–88) | I (*n* = 25) II (*n* = 7) III (*n* = 22) | THA with tantalum acetabular component (*n* = 58) | PrR (*n* = 43) PoR (*n* = 3) PoC (*n* = 23) | Unknown | 100% at final follow up | 96 (48–174) | Unknown |
| Rowell et al., 2019 (Australia) [13] | 46 (47) | 65 (29–84) | Unknown | Cemented THA with cup cage (*n* = 47) | PrR (*n* = 14) PoR (*n* = 43) | Unknown | 92% at 2 years, 81% at 4 years | Unknown (3–132) | Unknown |
| Wegrzyn, 2018 (France) [14] | 126 (131) | 64 (51–77) | I (*n* = 19) II (*n* = 63) III (*n* = 49) | Cemented dual mobility THA (*n* = 124) Cemented dual mobility THA with cup cage (*n* = 7) | PrR (*n* = 9) PrC (*n* = 83) | Unknown | Unknown | 33 (16–50) | Unknown |
| Erol et al., 2016 (Turkey) [15] | 16 (16) | 57 (28–73) | II (*n* = 7) III (*n* = 9) | Modified Harrington (*n* = 16) | PoR (*n* = 10) | 72 (56.6–90) | 75% at 12 months, 37.5% at 18 months | 21 (6–70) | 75% at 1 year 37.5% at 1.5 years |
| Bernthal et al., 2015 (USA) [16] | 50 (52) | 57 (23–88) | II (*n* = 24) III (*n* = 28) | Modified Harrington (*n* = 52) | Unknown | Unknown | 90.4% at 49 months | 24 (2–92) | Unknown |
| Tsagozis et al., 2015 (Sweden) [17] | 70 (70) | 64 (40–86) | II (22) III (40) Unknown (*n* = 8) | Cemented THA with cup cage (*n* = 70) | PrR (*n* = 11) PoR (*n* = 41) | Unknown | 92% at 1 year, 89% at 5 years | 12 (1–205) | 49% at 1 year 7% at 5 years |

**Table 1.** *Cont.*

| Study (Country) | Total no. Patients (Hips), *n* | Mean Age, Years (Range) | Harrington Class (*n*) | Reconstruction Technique (*n*) | Adjuvant Therapy (*n*) | Post op MSTS, % (Range) | Implant Survival | Mean Follow up, Months (Range) | Patient Survival Rate |
|---|---|---|---|---|---|---|---|---|---|
| Kiatisevi et al., 2015 (Thailand) [18] | 22 (22) | 54 (33–71) | II (*n* = 5) III (*n* = 17) | Cemented THA with cup cage (*n* = 19) Cemented THA (*n* = 3) | PoR (*n* = 22) | 70 (27–87) | 100% at final follow up | 8 (3–15) | 28% at 1 year |
| Shahid et al., 2014 (UK) [2] | 78 (78) | 61 (15–87) | Unknown | Modified Harrington (*n* = 35) Cemented THA (*n* = 32) Ice-cream cone prosthesis (*n* = 11) | PrR (*n* = 49) PrC (*n* = 47) | Unknown | Unknown | Unknown | 45% at 1 year 30% at 2 years 5% at 5 years |
| Vielgut et al., 2013 (Austria) [4] | 9 (9) | 62 (42–75) | II (*n* = 2) III (*n* = 6) IV (*n* = 1) | Modified Harrington (*n* = 9) | PoR (*n* = 4) PrR + PoR (*n* = 5) | Unknown | 100% at final follow up | 13 (2–30) | Unknown |
| Hoell et al., 2011 (Germany) [19] | 15 (15) | 62 (48–77) | II (*n* = 3) III (*n* = 12) | Cemented THA with cup cage (*n* = 15) | PrR (*n* = 10) PoR (*n* = 3) PrC (*n* = 11) | Unknown | 80% at final follow up | 14 (1–34) | Unknown |
| Khan et al., 2011 (Japan) [20] | 20 (20) | 60 (22–80) | I (*n* = 7) II (*n* = 3) III (*n* = 8) | Uncemented THA with tantalum acetabular component (*n* = 20) | PrR (*n* = 15) | Unknown | Unknown | 56 (26–85) | 45% at 1.5 years |
| Tang et al., 2011 (China) [21] | 15 (15) | 51 (20–71) | Unknown | Modular hemipelvic endoprosthesis (*n* = 15) | Unknown | 69.6 (20–90) | Unknown | 32 (19–60) | Unknown |
| Ho et al., 2010 (USA) [22] | 37 (37) | 63 (35–83) | III (*n* = 37) | Modified Harrington (*n* = 37) | PoC (*n* = 31) | 67 (30–87) | 71% at 1 year, 59% at 2 years, 49% at 5 years | 23 (0.5–112) | 63% at 1 year 55% at 2 years 39% at 5 years |
| Tillman et al., 2008 (UK) [23] | 19 (19) | 66 (48–83) | II (*n* = 6) III (*n* = 13) | Modified Harrington (*n* = 19) | Unknown | Unknown | 95% at final follow up | 25 (5–110) | Unknown |

PrR, pre-operative radiotherapy; PoR, post-operative radiotherapy; PrC, pre-operative chemotherapy; PoC, post-operative chemotherapy.

**Table 2.** Complications, *n* (%).

| Complication | Cemented THA (*n* = 35) | THA with Tantalum Acetabular Component (*n* = 115) | Cemented THA with Cup Cage (*n* = 151) | THA with Dual Mobility Liner (*n* = 131) | Modified Harrington (*n* = 374) | Hemipelvic Endoprosthesis (*n* = 63) | Reverse Ice-Cream Cone (*n* = 11) |
|---|---|---|---|---|---|---|---|
| Dislocation | - | 3 (2.6%) | 7 (4.6%) | 3 (2.3%) | 17 (4.9%) | 2 (13.3%) | 1 (9.1%) |
| Wound healing problem | - | - | - | - | 2 (0.5%) | - | - |
| Superficial infection | - | - | 1 (0.7%) | - | - | - | - |
| Deep infection | - | 5 (4.3%) | 2 (1.3%) | 4 (3%) | 13 (3.8%) | - | - |
| Aseptic loosening | 2 (5.7%) | - | 1 (0.7%) | - | - | - | - |
| Periprosthetic fracture | - | - | - | - | 3 (0.9%) | - | - |
| Metalwork failure | - | - | - | - | 6 (1.7%) | - | - |
| Pin migration | N/A | N/A | N/A | N/A | 3 (0.9%) | N/A | N/A |

### 3.3. Diagnoses and Adjuvant Therapy

The most common site of primary disease was breast (*n* = 230, 26.3%), followed by the lungs (*n* = 111, 12.7%), kidneys (*n* = 101, 11.5%), and prostate (*n* = 77, 8.8%).

The use of adjuvant therapy was commented on in 12 studies with 599 patients (68.5%) receiving pre- and/or post-operative chemotherapy and/or radiotherapy [2,9–15,17–19,22]. A total of 141 patients received neoadjuvant (pre-operative) chemotherapy, 235 patients received neoadjuvant radiotherapy, 55 patients received post-operative chemotherapy, and 159 patients received post-operative radiotherapy. A further 9 patients received a combination of these adjuvant therapies.

No other forms of adjuvant therapy were commented on in any of the included studies.

### 3.4. Total Hip Arthroplasty Alone

Total hip arthroplasty (THA) was carried out in 432 cases (49.1%). A total of 35 cases used THA alone [2,18], 151 cases used a flanged cup or cage [13,17–19], 131 cases used a dual mobility liner (seven of which also incorporated a cup cage) [14], and 115 cases used tantalum acetabular components [10,12,20].

All cases of THA alone utilized cemented components. A total of 3 were in patients with Harrington class II defects [18], and the remaining 32 in "smaller defects"; however the Harrington class was not specified [2]. Two patients developed aseptic loosening [2] with no other implant failures described.

THA with a cup cage was used for 27 patients with Harrington class II defects and 69 patients with Harrington class III defects. The Harrington class was not specified in the remaining 55 patients. Dislocations were reported in seven cases (4.6%), deep infection in two cases (1.3%) and aseptic loosening in one case (0.7%) [13,19].

A cemented dual mobility cup was used in 19 patients with Harrington class I defects, 63 patients with Harrington class II defects and 42 patients with Harrington class III defects. The use of a cemented dual mobility cup with an anti-protrusio cage was used in a further seven patients. Dislocation was reported in three cases (2.3%) and deep infection in four cases (3%) [14].

THA with tantalum acetabular components were utilised in 32 patients with Harrington class I defects, 10 patients with Harrington class II defects, 26 patients with Harrington class I or II defects, and 47 patients with Harrington class III defects. Dislocation occurred in three cases (2.6%) and deep infection in five cases (4.3%) [10,12,20].

### 3.5. Harrington Reconstruction

The Harrington procedure encompasses the use of threaded Steinmann pins inserted through the acetabular roof into the iliac wing with cement augmentation in addition to total

hip arthroplasty, with several modifications to this technique described. The Harrington procedure or modifications thereof were described in 374 cases (42.5%) [2,4,8–11,15–17,22,23].

The technique was carried out in 37 Harrington class I defects, 82 Harrington class II defects, 34 Harrington class I or II defects, 185 Harrington class III defects, and 1 Harrington class IV defect. Harrington classification was not specified in the remaining 35 cases.

There were 17 cases of dislocation (4.9%), 13 cases of deep infection (3.8%), 7 cases (2%) of metalwork failure, 6 cases (1.7%) of aseptic loosening, 3 cases (0.9%) of pin migration, and 3 (0.9%) periprosthetic fractures. Specific complications were not commented on in one study encompassing 30 patients [8].

### 3.6. Reverse Ice Cream Cone Prosthesis

A reverse ice cream cone prosthesis was used in 11 patients (1.25%) in cases with severe bone loss. There was one reported case of dislocation at one month post-operatively due to acetabular component anteversion, with no other prosthesis failures [2].

### 3.7. Modular Hemipelvic Endoprosthesis

A modular hemipelvic endoprosthesis was used in 63 patients (7.2%) [8,21]. In one study, they were used for Harrington class III lesions with bone destruction extending proximally to the inferior border of the sacroiliac joint (termed IIIb lesions) [8]. In this study, the prosthesis resulted in reduced incidence of complications, improved functional outcomes, and reduced local recurrence rates when compared to the Harrington procedure for similarly extensive Harrington class IIIb lesions. Exact complications from the prosthesis were not ascertained in this study.

A second study used a modular hemipelvic endoprosthesis for patients with a solitary periacetabular metastasis [21]. The tumour resection location was described by using the Enneking classification. Three patients had Type II (periacetabular), five patients had Type I/II (periacetabular and ilium), four patients had Type II/III (periacetabular and pubis), and three patients had Type I/II/III (whole hemipelvis) resection. There were two cases of dislocation (13.3%) and four patients (26.7%) had superficial wound healing problems. There were no cases of deep infection, aseptic loosening, or other mechanical failure.

### 3.8. Local Recurrence and Mortality Outcomes

Local recurrence of disease was reported in five studies only, ranging from 0.7 to 5.1% with a mean of 2.7%. Mortality outcomes are summarized in Table 1. The 1-year post-operative survival ranged from 28 to 75%, with the 5-year survival ranging from 5 to 39%. One study reported on 10-year mortality with a survival rate of 15% following either THA or a modified Harrington reconstruction [10].

### 3.9. Quality Assessment

All the included studies demonstrated a low to moderate risk of bias in all domains of the ROBINS-I tool, with an overall moderate risk of bias for all studies (Figure 2).

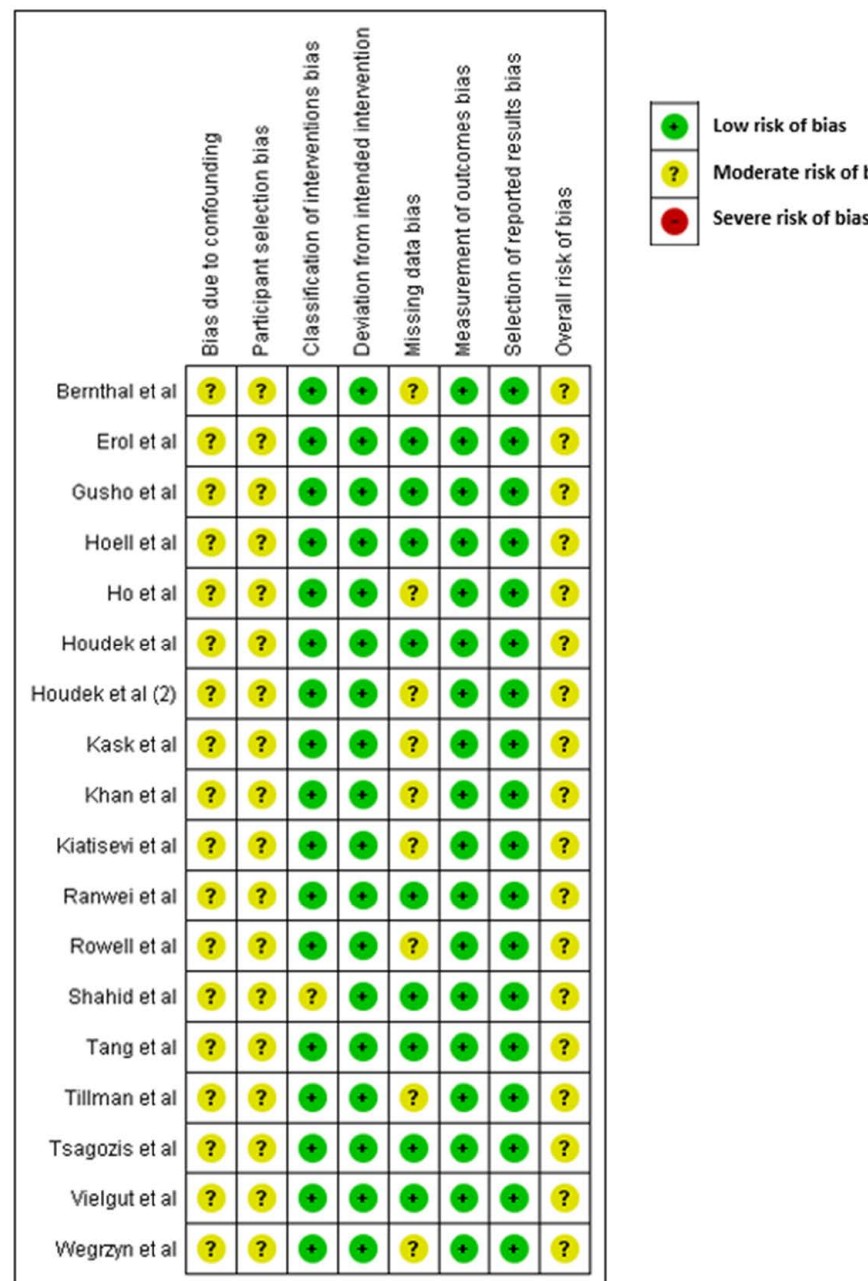

**Figure 2.** Results of quality assessment using ROBINS-I tool [2,4,8–15,17–23].

## 4. Discussion

In 1981, Harrington described a reconstruction technique for advanced periacetabular cancer secondary to metastatic disease [5]. Although there is still widespread use of the technique, over the past 40 years there have been significant advances in reconstructive options for such patients. Advances in imaging modalities such as CT scan and 3D reconstruction has permitted greater pre-operative planning. Technological advances have resulted in the development of more complex implants and modular endoprostheses. More recently, the advent of custom-made and 3D printing technology has been employed [24]. These novelties have allowed for the reconstruction of lesions with extensive bone loss, which would have previously resulted in poorer outcomes from hip arthroplasty with cement augmentation after the Harrington procedure. However, with more extensive procedures comes the added increase in morbidity from greater blood loss and larger surgical wounds [8].

Historically, the Harrington procedure was associated with high rates of dislocation and aseptic failure [25,26]. In this review of the contemporary literature, we found that the use of constrained and dual mobility liners has resulted in reduction of these complications. Bagsby et al. demonstrated no cases of dislocation or component failure in 68 patients that underwent a modified Harrington procedure with constrained liners in Harrington class II and III defects [27]. Wegrzyn et al. reported a dislocation rate of 2% from 126 patients that underwent reconstruction with a dual mobility cup with or without cement augmentation depending on the extent of the lesion [14]. The use of an antiprotrusio cage with or without the modified Harrington procedure has also been described to address larger defects. However, high dislocation rates (18.5%) were reported in Tsagozis et al.'s series of 70 patients that underwent a modified Harrington procedure with an antiprotrusio cage for Harrington class II and III defects [17].

The use of uncemented porous tantalum implants to address large acetabular defects has also been considered as an option for reconstruction. Tantalum has the theoretical advantages of high porosity allowing for extensive bony ingrowth as well as a low elastic modulus and high friction coefficient minimising the effect of stress shielding [28]. In Houdek et al.'s comparative study of total hip arthroplasty using either the Harrington technique (78 patients) or a tantalum acetabular reconstruction (37 patients), the 10-year cumulative incidence of acetabular component revision for loosening was 9.6% in the Harrington group versus 0% in the tantalum group, with no difference in functional outcomes [10], suggesting this may be a useful alternative in reconstruction of smaller defects.

The use of modern endoprostheses has become more common in large primary bone lesions. However, their application has also expanded to metastatic lesions associated with extensive bone loss. Wei et al. split Harrington class III lesions into class IIIa and IIIb, with the latter defined as bone destruction extending proximal to the inferior border of the sacroiliac joint. They subsequently compared the use of a modular hemipelvic endoprosthesis in 48 patients with these IIIb lesions compared to a cohort of six patients that underwent a modified Harrington procedure for such lesions. Their results demonstrated reduced surgical time, less intraoperative blood loss, improved functional outcomes and improved 2-year recurrence-free survival rates in the endoprosthesis group [8]. Furthermore, use of the reverse ice cone prosthesis resulted in satisfactory functional outcomes with minimal complications for cases large bone loss and pelvic discontinuity [2].

Newer technologies outside the scope of this review include the use of robotic-assisted reconstruction. One case report described a modified Harrington procedure with the use of robotic-assisted 3D navigation to aid with pin placement for a patient with metastatic renal cell carcinoma. They report satisfactory functional outcome and no complications or tumour recurrence at one year post-operatively [29]. This is the first report of this kind and may present a new avenue to improve patient outcomes.

Minimally invasive stabilization has also been described, with a recent study reporting on 38 patients with Harrington class II or III metastatic lesions managed with percutaneous screws and cement osteoplasty [30]. This resulted in significant improvement in pain and function with no occurrences of mechanical implant failure. Three patients (7.9%) went on to require repeat of the minimally invasive treatment and three (7.9%) required conversion to THA. This may be a promising alternative to be considered in the future with reduced morbidity when compared to more invasive procedures. However, selection of which patients would best benefit from a minimally invasive procedure are yet to be determined and warrants further research.

Limitations of this review include a lack of comparative data, with most studies reporting retrospective data. Hence, there is a risk of selection bias as well as inherent risk of confounding factors which may influence outcomes from the different reconstruction modalities.

## 5. Conclusions

This scoping review has provided an overview of the commonly used treatment modalities for metastatic periactebular lesions in the contemporary literature. Various advances in

imaging modalities and implant design have resulted in improved outcomes and allow for tailoring of the reconstructive procedure performed depending on the extent of the disease around the acetabulum. In future practice, there is scope for modular and custom-made endoprostheses to be used for more extensive lesions, and minimally invasive techniques may be possible in patients with smaller contained lesions. Further research is required with prospective, comparative, and ideally randomized data to establish which patients would benefit from the different reconstructive options. The best use of technologies such as robotic-assisted surgery to optimize outcomes also warrants further research.

**Author Contributions:** Conceptualization, S.K.N., T.A.K., O.S., K.V. and P.J.P.; methodology, S.K.N. and T.A.K.; formal analysis, S.K.N. and T.A.K.; investigation, S.K.N. and T.A.K.; resources, S.K.N. and T.A.K.; data curation, S.K.N. and T.A.K.; writing—original draft preparation, S.K.N. and T.A.K.; writing—review and editing, S.K.N., T.A.K., O.S., K.V. and P.J.P.; visualization, S.K.N. All authors have read and agreed to the published version of the manuscript.

**Funding:** This research received no external funding.

**Conflicts of Interest:** The authors declare no conflict of interest.

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
