# Peer review of "Outcomes of Hip Reconstruction for Metastatic Acetabular Lesions: A Scoping Review of the Literature"

_curroncol, doi:10.3390/curroncol29060307_

Round 1

Reviewer 1 Report

The topic is very interesting and the paper well written. I have a few concerns

Any other adjuvant therapy?eg embolization, elettrochemotherapy, criotherapy...

A table summarizing complications type and cumulative number (+ incidence) of complications in included studies would be an added value

Please provide a quality assessment of included studies

Discussion: which complication would fit better for every defect? Please discuss further

Author Response

There was no other mention of adjuvant therapies. This has now been clarified in the text (lines 114-115).

A table summarising complications has now been included (Table 2).

Quality assessment has now been carried out using the ROBINS-I tool as per Cochrane collaboration guidelines. This has been added to the Methods (lines 80-88) and Results (lines 178-180), as well as illustrated in Figure 2.

Reviewer 2 Report

Thank you for giving me the chance to revise this interesting paper.

Please, focus with more details on complications of different  reconstructions (a table with cumulative complications reported among included studies would be an added value)

In the discussion, a sort of flow chart suggesting which reconstruction would fit better for each defect would be very interesting

Author Response

A table summarising complications has now been included (Table 2).

Given the complex multifactorial decision-making when selecting a reconstruction method in these cases, the authors feel that universal guidance for management may not be clinically applicable. This review does aim, however, to provide a summary of the current reconstructive options, which we hope can help inform decision-making.

Reviewer 3 Report

The article does not bring any new information in the field of activity. A summary of the found works was practically made. I recommend checking the particularities of each case, their evolution, possible complications, postoperative death rate, either due to orthopedic causes or due to advanced neoplasm, what are the chances that a metastatic neoplasm, which was surgically operated, will recur in same place? These are interesting questions that definitely improve the article significantly.

Author Response

Information regarding local recurrence of disease and mortality has now been discussed in the text (lines 172-177). Furthermore, an extra column has been added to Table 1 with patient survival rate.

Reviewer 4 Report

Dear author, 

I consider the review is correct and precise. 
To gain clarity of data, I suggest reshaping the table by 'reconstruction technique' instead of 'study'. By doing so, the readers would have the whole picture of the findings regarding the number of patients, the treatment, and the outcomes. 

Author Response

The current Table 1 has been kept as is to incorporate feedback from other reviewers. However, a new table (Table 2) has also been added showing complications, which is structured according to reconstruction technique, showing the number of each performed.

Round 2

Reviewer 1 Report

The Authors made good efforts in the attempt to ameliorate their paper. It now merits publication

Reviewer 2 Report

The Authors made great efforts in the attempt to ameliorate their paper.

It is now suitable for publication

Reviewer 3 Report

The changes made add scientific value to the article.

Reviewer 4 Report

The manuscript is sufficiently improved for publication